# Resident Self-Tissue of Proinflammatory Cytokines Rather Than Their Systemic Levels Correlates with Development of Myelofibrosis in *Gata1*^low^ Mice

**DOI:** 10.3390/biom12020234

**Published:** 2022-01-30

**Authors:** Maria Zingariello, Paola Verachi, Francesca Gobbo, Fabrizio Martelli, Mario Falchi, Maria Mazzarini, Mauro Valeri, Giuseppe Sarli, Christian Marinaccio, Johanna Melo-Cardenas, John D. Crispino, Anna Rita Migliaccio

**Affiliations:** 1Department of Medicine, Campus Bio-Medico, 00128 Rome, Italy; m.zingariello@unicampus.it; 2Department of Biomedical and Neuromotorial Sciences, Alma Mater University, 40126 Bologna, Italy; paola.verachi@unibo.it (P.V.); fracesca.gobbo3@unibo.it (F.G.); maria.mazzarini4@unibo.it (M.M.); 3Department of Veterinary Medical Sciences, University of Bologna, 40126 Bologna, Italy; giuseppe.sarli@unibo.it; 4National Center for Drug Research and Evaluation, Istituto Superiore di Sanità, 00161 Rome, Italy; fabrizio.martelli@iss.it; 5National Center HIV/AIDS Research, Istituto Superiore di Sanità, 00161 Rome, Italy; mario.falchi@iss.it; 6Center for Animal Experimentation and Well-Being, Istituto Superiore di Sanità, 00161 Rome, Italy; mauro.valeri@iss.it; 7Dana-Farber Cancer Institute, Harvard Medical School, Boston, MA 02215, USA; christian_marinaccio@dfci.harvard.edu; 8Department of Hematology, St. Jude Children’s Research Hospital, Memphis, TN 38105, USA; johanna.melo-cardenas@stjude.org (J.M.-C.); john.crispino@stjude.org (J.D.C.); 9Altius Institute for Biomedical Sciences, Seattle, WA 98121, USA; 10Center for Integrated Biomedical Research, Campus Bio-Medico, 00128 Rome, Italy

**Keywords:** myelofibrosis, GATA1, proinflammatory cytokines, TGF-β1, interleukin 8, megakaryocytes, local microinflammatory states, systemic microinflammatory states

## Abstract

Serum levels of inflammatory cytokines are currently investigated as prognosis markers in myelofibrosis, the most severe Philadelphia-negative myeloproliferative neoplasm. We tested this hypothesis in the *Gata1*^low^ model of myelofibrosis. *Gata1*^low^ mice, and age-matched wild-type littermates, were analyzed before and after disease onset. We assessed cytokine serum levels by Luminex-bead-assay and ELISA, frequency and cytokine content of stromal cells by flow cytometry, and immunohistochemistry and bone marrow (BM) localization of GFP-tagged hematopoietic stem cells (HSC) by confocal microscopy. Differences in serum levels of 32 inflammatory-cytokines between prefibrotic and fibrotic *Gata1*^low^ mice and their wild-type littermates were modest. However, BM from fibrotic *Gata1*^low^ mice contained higher levels of lipocalin-2, CXCL1, and TGF-β1 than wild-type BM. Although frequencies of endothelial cells, mesenchymal cells, osteoblasts, and megakaryocytes were higher than normal in *Gata1*^low^ BM, the cells which expressed these cytokines the most were malignant megakaryocytes. This increased bioavailability of proinflammatory cytokines was associated with altered HSC localization: *Gata1*^low^ HSC were localized in the femur diaphysis in areas surrounded by microvessels, neo-bones, and megakaryocytes, while wild-type HSC were localized in the femur epiphysis around adipocytes. In conclusion, bioavailability of inflammatory cytokines in BM, rather than blood levels, possibly by reshaping the HSC niche, correlates with myelofibrosis in *Gata1*^low^ mice.

## 1. Introduction

The Philadelphia-negative myeloproliferative neoplasms (MPN) represent a continuum of diseases characterized by hyperproliferation of one or more blood lineages sustained by driver mutations in the thrombopoietin (TPO) axis (either in *MPL*, encoding the receptor for TPO, in *JAK2*, encoding the first element of the signal for MPL and other receptors of the cytokine superfamily, or in *Calreticulin*, encoding a chaperon protein that, when mutated constitutively, activates MPL) [1,2]. By a mechanism still poorly understood, MPN progress to myelofibrosis (MF), the most severe form of these diseases and to myeloid leukemia [3]. Since the progression of MPN to MF is not associated with the acquisition of specific second mutations [4], it has been hypothesized that it is driven by the establishment of a proinflammatory milieu in the bone marrow microenvironment which sustains fibrosis and hematopoietic failure in this organ and establishes hematopoiesis in extramedullary sites [5,6]. To test this hypothesis, the proinflammatory cytokines expressed at altered levels in myelofibrosis are the subject of extensive investigation, with the aim to identify and validate specific cytokine alterations that may be used as markers for prognosis of disease progression or targeted for patient-specific therapies. These studies are under the assumption that the level of a cytokine in the blood predicts its bioavailability in the bone marrow microenvironment and is fostered by the technical limitation that, due to the underlying fibrosis, bone marrow samples from MF patients suited for analyses are limited.

Previous studies have identified that the serum (and/or plasma) of patients with MF contains greater levels of several proinflammatory cytokines than that of nondiseased individuals. More specifically, the plasma of these patients contains greater levels of TGF-β, both as total and activated form [7], and interleukin-8 (IL-8), and the plasma levels of IL-8 were found to predict the severity of the symptoms and progression to leukemic transformation [8,9]. Both TGF-β and IL-8 are abnormally produced at high levels by the megakaryocytes present in the bone marrow of myelofibrosis patients [10,11]. However, these cytokines are also produced by monocytes, macrophages, and endothelial cells in response to proinflammatory stimuli [12,13]. In vitro studies have also shown that they can also be produced by other cell types present in the bone marrow microenvironment such as fibroblasts, neutrophils, and megakaryocytes [13,14,15,16,17,18]. The comparison of the production and response to these cytokines by normal and malignant megakaryocytes is of specific interest. The numerous malignant megakaryocytes found in the bone marrow of these patients are hypothesized as drivers of the fibrosis [19]. However, it is currently debated whether they drive the process by producing altered levels of inflammatory cytokines which activate other cell types (fibrocytes or fibrocytes possibly of macrophage origin) [20,21] to produce collagen and/or by responding to these cytokines by acquiring altered functions [22]. In fact, both TGF-β and IL-8 have been shown to increase the proliferation of megakaryocytes by retaining these cells in an immature state [11,22,23]. In addition, IL-8 has been reported to increase the interaction between megakaryocytes and neutrophils [24], the emperipolesis of which is thought to be responsible for increasing the release of TGF-β in the microenvironment [25]. These studies are of direct clinical interest, since preclinical data indicates that TGF-β trap and IL-8 inhibitors rescue the myelofibrosis phenotype in animal models [11,25], and clinical trials with these drugs are currently under investigation [26].

In the present study, we clarify the relationship between the serum levels of proinflammatory cytokines and their bioavailability in the bone marrow microenvironment using the well-characterized *Gata1*^low^ mouse model of myelofibrosis [21]. We also investigate the cell populations in the bone marrow microenvironment which are responsible for their increased bioavailability and the consequences of their alterations which, in addition to fibrosis, include a profound reshaping of the microenvironment architecture that alters the distribution of the hematopoietic stem cells within the bone marrow architecture. 

## 2. Materials and Methods

### 2.1. Mice

Transgenic mice were bred in the animal facility of Istituto Superiore di Sanità as described [27,28]. Littermates were genotyped at birth by PCR [28] and those found not to carry the expected mutation(s) were used as wild-type controls. We analyzed the following transgenic lines: *Gata1*^low^ mice, mice lacking *P-selectin* (*Psel*^null^) [29], and double *Gata1*^low^*Psel*^null^ mice [30]. To study the HSC localization within the bone marrow architecture, we analyzed double *huCD34tTA/TetO-H2BGFP* mice carrying the *histone H2B* gene fused with *GFP* under the control of the regulatory sequences of *human CD34* which in mice are active only in hematopoietic stem cells (HSC) [31,32] (defined as *huCD34-GFPH2B* from now on) and triple *huCD34tTA/TetO-H2BGFP/Gata1*^low^ mice (defined *huCD34-GFPH2B*/*Gata1*^low^*)* generated according to standard genetic approaches in the animal facility of Istituto Superiore di Sanità. All the mutations had been carried in the CD1 background for more than 10 generations. Mice were divided into two groups based on age: 5–7 months (adult) and 12–14 months (old) group, unless otherwise indicated. In selected experiments, wild-type and *Gata1*^low^ littermates were treated with SB431542 (cat# S4317-5GM, Sigma Aldrich, Saint Louis, MI, USA), an inhibitor of the first tyrosine kinase of the TGF-β1 receptor type I signaling [33]. In these experiments, 12-month-old mice (12 mice/experiment) were intraperitoneally injected with SB431542 (60 µg/kg/day) or vehicle (same volume) for 1 month. Wild-type (WT) CD1, DBA2, and C57BL6 mice were purchased from Charles River (Calco, Lecco, Italy). Gender was considered as an independent variable and results obtained with females and males were combined because they were found not to be statistically different.

### 2.2. Blood Collection

Blood was collected from the retro-orbital plexus into ethylen-diamino-tetracetic acid-coated microcapillary tubes (20–40 µL/sampling) from mice that had previously topically anesthetized with lidocaine (one drop/eye) (EDRA S.p.A., Milan, Italy). Serum was prepared by allowing blood clotting for 2 h at room temperature and centrifugation for 20 min at 2000× *g*. Serum samples were stored in aliquots at −80 °C, thawed once, and analyzed within 20 min from thawing. Serum from 5–7 months (adult) and 12–14 months (old) wild-type CD1, C57BL-6, and DBA2 mice were purchased from Charles River Laboratories.

### 2.3. Cytokine Profiling

The serum was spun at 10,000× *g* for 10 min at 4 °C to remove particulates. Cytokine levels were quantified using the mouse cytokine/chemokine magnetic bead panel for 32 cytokines from Merck Millipore (cat# MCYTMAG70PMX32BK, Burlington, MA, USA) according to manufacturer’s instructions. Data were acquired using a Luminex^®^ 200™, and cytokine concentrations were calculated using xPONENT^®^ software (xPONENT LX100/LX200 v3.1., Luminex Corporation, Auxin, TX, USA) against a standard curve.

### 2.4. Enzyme-Linked Immunosorbent Assay (ELISA)

Serum levels of lipocalin-2 (LCN2), CXCL1 and TGF-β1 were determined in duplicate with the Quantikine ELISA Kits (cat# MLCN20, MKC00B and DB100B, respectively, R&D Systems, Minneapolis, MN, USA), as described by the manufacturer. Optical densities were determined using a microplate reader set to 450 nm (VICTOR^®^ Nivo™, Perkin Elmer, Milan, Italy).

### 2.5. Flow Cytometry Determination of Microenvironmental Cells and HSC

The frequency of endothelial cells (ECs), mesenchymal stem cells (MSCs), and osteoblasts (OBCs) was determined by analyzing hematopoietic cell-depleted bone marrow cells obtained by carefully crushing the femurs in Iscove’s Modified Dulbecco’s Medium according to flow cytometry criteria defined by the Passegué Laboratory [34]. Briefly, ECs were defined as Lin^−^/CD45^−^/CD31^+^/Sca-1^+^ cells, MSCs as Lin^−^/CD45^−^/CD31^−^/CD51^+^/Sca-1^+^ cells, and OBCs as Lin^−^/CD45^−^/CD31^−^/CD51^+^/Sca-1^−^ cells. The frequency of megakaryocytes (MKs) and neutrophils (Neu) was instead determined by analyzing bone marrow cells labeled with FITC-A CD61/PE-A CD41 or PE-A Gr1/PE-Cy7-A CD11b, as described [33,35]. For HSC determinations, mononuclear bone marrow cells were incubated with a cocktail of the following antibodies: APC-CD117, APC-Cy7-Sca1, PE-Cy7-CD150, biotin-labeled anti-mouse CD48, and biotin-labeled anti-lineage antibodies. After 30 min of incubation on ice, cells were washed once and stained with streptavidin-PE-Cy5, and cell fluorescence was analyzed with a Gallios analyzer (Beckman Coulter, Brea, CA, USA). The total HSC population was defined as LSK (Lin^−^/CD48^−^/c-kit^+^/Sca-1^+^), while long-term repopulating HSC were defined by the SLAM phenotype (Lin^−^/CD48^−^/c-kit^+^/Sca-1^+^/CD150^+^) as previously described [30,36]. Nonspecific signals and dead cells were excluded by appropriate fluorochrome-conjugated isotype and propidium iodide staining, respectively. All the antibodies were purchased from BD-Pharmingen (San Diego, CA, USA). Fluorescent signals were measured with the FACS Aria (Becton Dickinson, Franklin Lakes, NJ, USA), and dead cells were excluded by Sytox Blue staining (0.002 mM, Molecular Probes, Eugene, OR, USA). Results were analyzed with the Kaluza analysis version 2.1 (Beckman Coulter). Examples of the gates used to identify the stromal cells and the HSC are presented in Appendix A.

### 2.6. Histological and Immunohistochemical Analyses

Femurs were fixed in formaldehyde (10% *v*/*v* with neutral buffer), treated for 1 h with bone marrow biopsy decalcifying solution (Osteodec; Bio-Optica, Milan, Italy), and included in paraffin. Sections (3 μm) were stained either with hematoxylin–eosin (H&E; Hematoxylin cat# 01HEMH2500; Eosin cat# 01EOY101000; Histo-Line Laboratories, Milan, Italy) or reticulin (both from Bio-Optica, Milan, Italy) or subjected to immunohistochemistry with anti-TGF-β1 (cat# sc-130348, Santa Cruz Biotechnology, Santa Cruz, CA, USA), anti-LCN2 (cat# MBS178180, MyBioSource, San Diego, CA, USA), anti-CXCL1 (cat# ab86436, Abcam, Cambridge, UK), anti-CXCR1 (cat# GTX100389, Genetex, Irvine, CA, USA), anti-CXCR2 (cat# catalog ab14935, Abcam), and anti-BMP-4 (cat# ab235114, Abcam) antibodies. Immunoreactions were detected with avidin-biotin immunoperoxidase (Vectastain Elite ABC Kit, Vector Laboratories, Burlingame, CA, USA) and the chromogen 3,3′-diaminobenzidine (0.05% *w*/*v*, cat# ACB999, Histo-Line Laboratories). Slides were counterstained with Papanicolaou’s hematoxylin (Histo-Line Laboratories). Images were acquired with an optical microscope (Eclipse E600; Nikon, Shinjuku, Japan) equipped with the Imaging Source “33” Series USB 3.0 Camera (cat# DFK 33UX264; Bremen, Germany). Reticulin fibers were quantified on five different areas/femur/mouse from at least four mice per group using the ImageJ program (version 1.52t) (National Institutes of Health, Bethesda, MD, USA) (see Appendix A). The level of the expressions of LCN2, TGF-β1, CXCL1, CXCR1, and CXCR2 per cell was also evaluated with the ImageJ program by counting the number of cells that exceeded the intensity set as threshold (see Appendix A).

### 2.7. Confocal Microscopy Determinations

Femurs were fixed, decalcified, and included in paraffin, as described above. Sections (5 μm) were stained with DAPI (D9542-5MG, Sigma Aldrich) and analyzed with the confocal microscope Zeiss LSM 900 (Carl Zeiss GmbH, Jena, Germany) in the Airyscan mode. Excitation light was obtained by a Laser Dapi (405 nm) for DAPI and the argon ion laser (488 nm) for GFP. Optical thickness varied from 0.50 μm for the 20× objective to 0.20 μm for the 63× objective. Images were process and analyzed with the Zen Blue (3.2) software (Carl Zeiss GmbH, November 2021) and the ImageJ (version 1.53t) software (National Institutes of Health http://imagej.nih.gov/ij, accessed on 23 November 2018) (see movie 1 and 2, for detail). Three-dimensional reconstructions were obtained by the full set of stack images, 15 images for the 20× objective and 34 images for 63× objective, by the Zen Blue software.

### 2.8. Expression Profiling

For microarray analyses, total RNA was extracted from the bone marrow and spleen of 8- to 10-month-old *Gata1*^low^ and wild-type littermates, purified with Rneasy Mini Kit (Qiagen, Germantown, MD, USA) and hybridized to the Illumina Mouse WG-6V2 Bead Chip gene expression array as described [37]. Functional annotation clustering was performed with the David Bioinformatic Database (David Bioinformatics Resources 6.7 NIAID/NIH). Microarray data have been deposited in the Gene Expression Omnibus database (GSE89630) https://www.ncbi.nlm.nih.gov/geo/query/acc.cgi?token=ijgpwgsyjhwpdoj&acc=GSE89630 [37]. Gene set enrichment analysis (GSEA) was performed with the GSEA application version 4.1.0 (https://www.gsea-msigdb.org/gsea/index.jsp) with gene set setting for the permutation parameter and Signal2Noise for the metric parameter [38,39].

### 2.9. Statistical Analyses

All the data have a normal distribution, as assessed by Shapiro and Wilk’s W test (Graph Pad Software, La Jolla, CA, USA) and are presented either as mean (±SD) or as median (min–max) of at least three separate experiments, as indicated. Statistical analyses among groups were performed either by *T*-test (comparisons among two groups) or by Tukey for multiple comparison test (comparisons among more than three groups), as appropriate. Differences among groups were considered statistically significant with a *p* < 0.05.

## 3. Results

### 3.1. Modest Differences in the Cytokine Profile of Gata1^low^ Mice with Age

The levels of 32 cytokines in the serum from *Gata1*^low^ mice at 5–7 (when they have not developed myelofibrosis) and 12–14 (with bone marrow fibrosis) months of age were studied. Of the 32 cytokines analyses, only 18 were present at detectable levels in the serum of the mice (IL-6, Eotaxin, IL1a, IL5, IL12p70, IL12p40, IL13, LIX, IL9, G-CSF, IP10, KC (also known as CXCL1), MCP1, MIP1a, MIP1b, MIG, RANTES, TNFα) while the concentrations of 14 cytokines (GM-CSF, IFNγ, IL2, IL4, IL10, IL3, IL7, LIF, VEGF, IL1β, IL15, IL17, and M-CSF) were below the levels detectable with this assay (3.2 pg/mL). Detectable cytokines are presented in Figure 1A,B. Results were compared with those observed in CD1 mice of comparable age and with 12–14-month *Gata1*^low^ mice lacking the *P-selectin* gene (*Psel*^null^), as negative control. Both wild-type mice and double *Gata1*^low^*Psel*^null^ mice do not develop fibrosis in bone marrow [27,28,30]. The results showed changes in a few cytokines with mostly decreased levels of LIX, IL1a, and Eotaxin in young and old *Gata1*^low^ animals, and decreased levels of IL12p70 in old *Gata1*^low^ mice compared to wild-type controls.

Given the great interest in TGF-β1, LCN2, and CXCL1 in the pathobiology of myelofibrosis [8,40,41], the serum levels of these three inflammatory cytokines were compared by ELISA using larger cohorts of mice. In addition, in this set of experiments, *Gata1*^low^*Psel*^null^ (and *Psel*^null^ mice) were analyzed as additional negative control. Age and sex did not affect the values observed in the different experimental groups. Therefore, results obtained with young and old, or males and females, were combined. The serum of *Gata1*^low^ mice contained levels of LCN2 and (on this large cohort of mice) also of CXCL1 significantly lower than that of CD1 mice or of mice lacking *Psel* with or without the *Gata1*^low^ mutations (Figure 1C). In a previous publication, we demonstrated that the serum of *Gata1*^low^ mice contains TGF-β1 levels modestly (two-fold) higher that of CD1 mice (2.4 vs. 1.5 ng/mL, respectively) [33]. Here, we confirm that these levels are slightly higher also than those of *Psel*^null^ and *Gata1*^low^*Psel*^null^ mice. 

### 3.2. The Genetic Background Is a Confounding Factor in Determining the Normal Levels of TGF-β1, LCN2 and CXCL1 in Mice

Given the great importance of the proinflammatory cytokines in the development of myelofibrosis [5,6], the modest differences found in the serum levels among CD1 mice harboring the *Gata1*^low^ mutation or wild-type at this locus was puzzling. It has been previously shown that the mouse strain plays an important role in the manifestation of myelofibrosis induced by the *Gata1*^low^ mutation [28]. In the C57BL/6 background, the *Gata1*^low^ mutation is lethal at birth with the few (~5%) surviving mice developing myelofibrosis within the first 1–2 months [42,43]. In the CD1 background, *Gata1*^low^ mice are viable at birth because they rapidly recruit the spleen as extramedullary site but slowly develop myelofibrosis while aging [27,28]. By contrast, in the DBA/2 background, *Gata1*^low^ mice are born viable and develop thrombocytopenia and anemia with age but never develop myelofibrosis [28].

A careful analysis of the pathobiology of laboratory mice with age and of the cause of their natural death indicates the existence of a vast variegation of inflammatory manifestations among different strains [44]. In fact, in addition to neoplasms of the hematopoietic, lung, liver, and mammary glands, which are developed by mice of most strains, CD1 mice die more frequently of pathologies associated with inflammatory signatures such as amyloidosis, arteritis, glomerulonephritis, ulcerative dermatitis, fatty liver, and pneumonia than C57BL/6 and DBA/2 mice [44]. These results suggest that the absence of differences between the levels of cytokines present in the serum between *Gata1*^low^ and wild-type littermates described in Figure 1 may be confounded by the fact that the serum of mice in the CD1 background contains basal levels of proinflammatory cytokines higher than that of other strains. To test this hypothesis, we compared the levels of TGF-β1, LCN2, and CXCL1 in the sera from CD1, C57BL/6, and DBA2 mice obtained from commercial sources (Figure 2). Serum was collected from both males and females at 5–7 months (adult) and 12–14 months (old) of age and since the results from females and males or young and adult mice were similar, they were pooled. Indeed, there was a good correlation between the reported manifestation of inflammatory diseases [44] and serum levels of TGF-β1, LCN2, and CXCL1 among different strains. CD1 mice expressed levels of all three inflammatory cytokines significantly greater than those expressed by DBA/2 mice, while C57BL/6 mice expressed only levels of LCN2 and CXCL1 greater than DBA/2 mice.

To assess whether the wild-type littermates of *Gata1*^low^ mice develop with age pathological conditions associated with inflammation, we conducted histopathological examinations of sections from their organs (Figure 3). Hematoxylin/eosin-stained sections of lungs, hearts, kidneys, spleens, livers, and skins from six mice (four males and two females) below 6 months old and from 15 mice (nine males and six females) above 11 months of age were analyzed (Figure 3, and data not shown). None of the mice younger than 6 months showed detectable changes in the tissues observed (data not shown) while chronic inflammation, often involving more than one organ, was often apparent in samples from older animals. The most frequent lesion was interstitial pneumonia (7/15 mice) (Figure 3A,B), followed by nephritis (5/15 mice), with a mixed expression of chronic glomerulonephritis associated with interstitial nephritis (Figure 3C). Arteriopathy with mural calcification was apparent in kidneys of two mice (Figure 3D), while only in one mouse was amyloidosis in the spleen (Figure 3E), interstitial hepatitis (Figure 3F), chronic dermatitis (Figure 3G) and steatonecrosis (Figure 3H) observed. The mouse with amyloidosis showed interstitial hepatitis as associated lesion.

### 3.3. The Bone Marrow Microenvironment of Gata1^low^ Mice Contain Abnormally High Levels of Megakaryocytes, Endothelial Cells, Mesenchymal Cells, and Osteoblasts

HSC are sustained by specific niche cells of the bone marrow microenvironment, namely, by endothelial cells present in the microvasculature [45]. Evidence is emerging, however, that the nature of the supporting niche changes with age and/or under inflammatory states, although the identity of the cells supporting the HSC under these conditions has not been clarified as yet [46,47]. In addition, MPN is associated with profound differences in the composition of the cells of the microenvironment. The Passegué laboratory had demonstrated that the malignant hematopoietic cells of the Bcr/Abl-driven mouse model of chronic MPN progressively remodels the endosteal BM niche into a self-reinforcing leukemic niche that impairs normal hematopoiesis, favoring the leukemic stem cell function and contributing to bone marrow fibrosis [34]. This remodeling requires direct cell–cell interaction and changes in the expression levels of TGF-β, Notch, and other inflammatory signaling. In addition, in the case of JAK2-driven MPN, it has been shown that in patients and in animal models, the disease is driven by a consistent reduction in the numbers of sympathetic nerve fibers, supporting Schwann cells, which regulates the HSC supporting functions of a subpopulation of nestin^+^ mesenchymal stem cells [48,49].

To clarify the cell population(s) responsible for establishing a proinflammatory milieu in the bone marrow of *Gata1*^low^ mice, we first compared the frequency of stromal cells, and of hematopoietic cells (megakaryocytes and neutrophils), with microenvironmental shaping functions [50,51] in the bone marrow from *Gata1*^low^ and wild-type littermates (Figure 4).

The frequencies of megakaryocytes, endothelial and mesenchymal cells, and osteoblasts in the bone marrow from *Gata1*^low^ mice are all greater than normal, while those of the neutrophils are within normal ranges (Figure 4). The greater frequency of megakaryocytes is consistent with previous reports [52] while that of endothelial cells and of osteoblasts is consistent with the increased vascularization and osteogenesis observed in these mice [52,53], which recapitulate that seen in myelofibrosis patients [54]. Deletion of *Psel* and pharmacological inhibition of TGF-β normalized the frequency of endothelial cells, of mesenchymal cells, and osteoblasts found in *Gata1*^low^ bone marrow. These observations are consistent with the finding that deletion of *Psel*, probably by reducing TGF-β bioavailability, and inhibition of TGF-β both reduces osteopetrosis and neovascularization in *Gata1*^low^ mice [30,33,55].

These results indicate that the microenvironment of the bone marrow from *Gata1*^low^ mice contains altered levels of stromal and hematopoietic cells which can all be targets and/or producers of proinflammatory cytokines.

### 3.4. The Bone Marrow from Gata1^low^ Mice Contains High Levels of TGF-β1, LCN2, and CXCL1 Produced by the Malignant Megakaryocytes

By contrast with the modest differences in proinflammatory cytokines in the serum of *Gata1*^low^ mice and wild-type littermates (Figure 1C), in previous publications, we reported immunohistochemistry studies indicating that the bone marrow from *Gata1*^low^ mice contains levels of TGF-β1, LCN2, and CXCL1 greater than normal ([11,33] and Appendix A). These cytokines may be produced by several cell types present in the bone marrow microenvironment: TGF-β1 may be produced by endothelial cells [56] and megakaryocytes [33], LCN2 by neutrophils [57], and CXCL1 by megakaryocytes, endothelial cells, osteoblasts, and neutrophils [24,58].

The increased frequencies of stromal cells described above raises, then, the question of whether the increased cytokine bioavailability in the bone marrow of *Gata1*^low^ mice is a consequence of altered numbers of producing cells and/or by altered levels of cytokine produced by a specific cell type. To answer this question, we compared the levels of cytokine expressed by individual cell types identified on the basis of morphological criteria in bone marrow sections immune-stained with antibodies against the three different cytokines. Endothelial cells were recognized as flat cells surrounding structures resembling vessels (i.e., optically empty channels containing red blood cells); osteoblasts were identified as cuboidal cells on the endosteum of the bones; megakaryocytes on the basis of their size (10 times greater than that of any other cell type) and the polylobate morphology of their nuclei; and neutrophils on the basis of their small size (9 μm) and kidney-shaped nuclei (Figure 5).

As expected [33,56], in the bone marrow of wild-type mice, TGF-β1 was expressed at detectable levels by endothelial cells and megakaryocytes (Figure 5A). In addition, in the bone marrow from *Gata1*^low^ mice, TGF-β1 was expressed by endothelial cells and megakaryocytes but, as previously reported [33], the TGF-β1 immunostaining of the malignant megakaryocytes is significantly stronger than that of the corresponding wild-type cells. These results suggest that the increased bioavailability of TGF-β1 in the bone marrow microenvironment of the mutant mice is due both to the increased number of megakaryocytes and to the greater amount of TGF-β1 they produce at the single cell level (Figure 5B).

As expected [57], in the bone marrow of wild-type mice, LCN2 is contained mostly in neutrophils (Figure 5A). By contrast, in the bone marrow of *Gata1*^low^ mice, LCN2 is expressed at detectable levels both by neutrophils and megakaryocytes (Figure 5B). The expression of LCN2 in the malignant megakaryocytes is consistent with the great levels of LCN2 previously reported in megakaryocytes expanded in vitro from MF patients [59]. Therefore, the increased bioavailability of LCN2 in the microenvironment of *Gata1*^low^ mice is due to production by a novel cell type, i.e., the malignant megakaryocytes. However, we may not formally exclude that the source of LCN2 in the *Gata1*^low^ megakaryocytes is not neutrophils engulfed by emperipolesis in these cells. Further studies are necessary to clarify this point.

As expected [24,58], in the bone marrow of wild-type mice, CXCL1 is expressed at detectable levels by endothelial cells, osteoblasts, megakaryocytes, and neutrophils (Figure 5A). In the bone marrow of *Gata1*^low^ mice, CXCL1 is expressed by the same cells which express this growth factor in the wild-type organ but the expression in the malignant megakaryocytes is much greater than in their corresponding wild-type cells (Figure 5B). Therefore, as for TGF-β1, the increased bioavailability of CXCL1 in the microenvironment of *Gata1*^low^ mice is due to the increased numbers of megakaryocytes that produce greater levels of this growth factor per cell.

### 3.5. Megakaryocytes from the Bone Marrow of Gata1^low^ Mice Express Greater Levels of CXCR1 and CXCR2

CXCL1 has been described to exert its biological effects by activating CXCR1 expressed by endothelial cells, osteoblasts, and megakaryocytes [60,61,62,63], while CXCR2 is specifically expressed by neutrophils [58,63]. To clarify which cell type may be responding to the greater levels of CXCL1 observed in the bone marrow of *Gata1*^low^ mice, immunostaining studies with anti-CXCR1 and CXCR2 antibodies were performed (Figure 6). In wild-type bone marrow, CXCR1 expression was detectable at great levels on endothelial cells and neutrophils and at lower levels on osteoblasts and megakaryocytes, while CXCR2 was detectable at high levels on neutrophils, at low levels on endothelial cells and megakaryocytes, and levels barely detectable on osteoblasts. By contrast, as previously reported [11], the bone marrow from *Gata1*^low^ mice expressed total levels of CXCR1 greater than normal (Appendix A). At the single cell levels, CXCR1 and CXCR2 were expressed at levels similar to those found in wild-type mice in the endothelial cells, osteoblasts, and neutrophils of the mutant mice, but it was expressed at levels much greater than normal in megakaryocytes (Figure 6), indicating that the great content of receptors observed in the bone marrow is likely a reflection of its great megakaryocyte content.

In previous publications, we have demonstrated that increased TGF-β bioavailability activates a TGF-β expression signature in the bone marrow from *Gata1*^low^ mice [33]. Since LCN2 activates CXCL1 expression [64], the greater levels of CXCL1 observed in the bone marrow of the mutant mice (Figure 5 and Appendix A) provides indirect proof that the LCN2 signaling is also activated in this organ. However, there is no indication so far that increased CXCL1 bioavailability activates this pathway in the bone marrow from *Gata1*^low^ mice. To fill this gap, we combed the published expression profile of *Gata1*^low^ bone marrow [65] for genes downstream to the two receptors of CXCL1, CXCR1, and CXCR2 (Figure 7). This analysis demonstrated a statistically significant activation of genes downstream to both CXCR1 and CXCR2, supporting our hypothesis that the proinflammatory signature of the bone marrow from *Gata1*^low^ mice includes activation of CXCL1 and CXCL2.

### 3.6. In the Bone Marrow from Old Wild-Type Mice, HSC Are Distributed around Adipocytes, While in That from Gata1^low^ Mice, HSC Are Localized in Areas of the Medulla between Bone Trabeculae and Vessels Which Contain Great Numbers of Megakaryocytes

The *huCD34-GFPH2B* mice were generated by the Moore Lemischka Laboratory to track the changes occurring in HSC properties when these cells divide [32]. Since the expression of *GFPH2B* is driven by the regulatory sequences of the human *CD34* gene, which in mice are active only in HSC [31], the *GFPH2B* transgene is expressed only by HSC with the stringent SLAM phenotype, which correspond to cells with long-term repopulation potential [32]. Since the GFPH2B protein is a histone incorporated in the chromosomes in stoichiometrically constant ration with the amount of their DNA, the levels of GFP expressed by the HSC in each mouse are extremely constant. However, since h*CD34* is no longer active as soon as the progeny of the HSC are committed to differentiate, only the immediate committed HSC progeny (short-term repopulating HSC and multilineage progenitor cells) retain some GFP signal inherited from the chromatids of their mother [32]. Studies on prospective cell isolation based on levels of GFP expression combined with functional analyses have demonstrated that only long-term repopulating HSC are GFP^high^ while short term repopulating HSC are GFP^medium^, multipotent progenitor cells are GFP^dim^, and lineage-restricted progenitor cells no longer express detectable levels of GFP [32].

By flow cytometry, we confirmed that GFP is not expressed by the cKIT^−^ cell fraction, which does not contain HSC/progenitor cells of the bone marrow from *huCD34-GFPH2B* mice, while it is expressed over a good range of intensities by the LSK cells with the greatest levels expressed by the 18% of LSK with the SLAM phenotype, which identified the long-term repopulating HSC (Figure 8). To our surprise, however, GFP was expressed by the cKIT^−^ cells from the bone marrow of *huCD34-GFPH2B**/Gata1*^low^ mice although it is expressed over a good range of intensity by LSK with the greatest levels detected in very few (1%) LSK with a SLAM phenotype (Figure 8). The reduced numbers of SLAM cells detected in the bone marrow of *huCD34-GFPH2B/Gata1*^low^ mice are consistent with previous publications which have demonstrated that in *Gata1*^low^ mice, most of the SLAM cells are in the spleen [30].

By confocal microscopy, cells expressing GFP were clearly detectable in the bone marrow from both *huCD34-GFPH2B* and *huCD34-GFPH2B/Gata1*^low^ littermates (Figure 9 and Figure 10). Stacking analyses indicated that in wild-type mice, GFP was mostly detectable in small cells where it was localized in the nucleus and in very few large cells where it was localized in the cytoplasm (Appendix A). GFP was mostly localized in the nucleus of small cells also in the bone marrow from *huCD34-GFPH2B/Gata1*^low^ mice but, in contrast with *huCD34-GFPH2B* mice, in this case the GFP was localized in the cytoplasm of a great number of large cells that had the morphology of macrophages (Appendix A, and data not shown). These results suggest that the large proportion of *huCD34-GFPH2B/Gata1*^low^ cKIT^−^ bone marrow cells that had been found to express GFP by flow cytometry (Figure 8) are likely macrophages that have phagocytized the malignant HSC which had likely died by apoptosis induced by the proinflammatory environment of the *huCD34-GFPH2B/Gata1*^low^ bone marrow [66].

Using these data as a foundation, we determined the localization of small GFP-tagged cells (which represent long- and short-term repopulating HSC and multilineage progenitor cells) within the bone marrow architecture of *huCD34-GFPH2B* and *huCD34-GFPH2B/Gata1*^low^ mice (Figure 9 and Figure 10). In the femur from old wild-type mice, GFP-tagged cells were localized either in clusters in the epiphysis of the femur or in the diaphysis area within the medulla (Figure 9). The brightest GFP-cells were found in the epiphysis and lined optically empty circles, a feature that characterizes areas occupied by adipocytes (Figure 9 and Appendix A). Interestingly, the cells around each cluster expressed homogenous level of GFP intensity (either high or low), suggesting that in old *huCD34-GFPH2B* mice cells, HSC with similar potency tend to remain localized together.

In agreement with the flow cytometry data indicating that in the bone marrow of *huCD34-GFPH2B/Gata1*^low^ mice GFP^high^ SLAM cells are very few, the spleen from *Gata1*^low^ mice contained much more GFP+ cell than that from *huCD34-GFPH2B* mice (data not shown). In addition, in huCD34-*GFPH2B/Gata1*^low^ mice, the nuclear fluorescent intensity in the cells from the spleen was significantly greater than that from the cells of the bone marrow (17.85 ± 12.98 vs. 6.88 ± 4.47 mean fluorescent intensity/cell in the spleen and in bone marrow, respectively). These data confirm previous observations indicating that in *Gata1*^low^ mice, a vast majority of the SLAM are in the spleen [30].

By contrast with the localization of GFP cells observed in the bone marrow from *huCD34-GFPH2B* mice, although the femurs of *huCD34-GFPH2B/**Gata1*^low^ mice contain more adipocytes than those of their *huCD34-GFPH2B* littermates (Figure 10 and data not shown), cells expressing GFP were rarely associated with the optically empty circles but instead were localized mostly in the medulla in areas containing microvessels and neoformed bone trabeculae (Figure 10). These areas also contained megakaryocytes.

Angiogenesis and neo-bone formation is sustained by the interplay between TGF-β, VEGF, and BMP-4 [67,68,69,70]. We have previously shown that the bone marrow of *Gata1*^low^ mice contains high levels of VEGF [52] and TGF-β [33]. We show here that this bone marrow also expresses high levels of BMP4, especially at the levels of the megakaryocytes (Figure 11). In a previous publication, it has been shown that megakaryocytes are responsible for the osteopetrosis observed in *Gata1*^low^ mice by secreting the bone matrix proteins osteonectin, bone sialoprotein, and osteopontin [53]. It is possible, then, that the megakaryocytes present within the cluster which engulf the GFP+ cells are releasing the growth factors (TGF-β, VEGF, and BMP-4) which, by inducing neo-angiogenesis and bone formation, are shaping the microenvironment of the *Gata1*^low^ femur, increasing the niches for short-term repopulation HSC and multilineage progenitor cells. These cells, however, are induced into apoptosis by the proinflammatory milieu of this microenvironment, resulting in hematopoietic failure.

## 4. Discussion

Numerous studies are currently analyzing the changes occurring at the level of HSC with aging under the assumption that these changes may predispose to the development of myeloproliferative disorders and possibly to leukemia. Aging affects the HSC directly, by inducing epigenetic changes, and indirectly, by altering the supportive role of the microenvironment [46,47,71]. These effects are supposed to be mediated by proinflammatory cytokines produced by the organism in response to environmental insults [72]. The inflammatory, aging, and HSC alteration circuit is usually investigated using animal models harboring loss or gain of function mutations in key genes. Here, we elucidated whether the proinflammatory milieu of the bone marrow microenvironment is correctly reflected by the cytokine content in the blood and the effects of this milieu on the development of myelofibrosis and of bone marrow failure using a mouse strain naturally predisposed to develop pathologies driven by inflammation.

We first demonstrated that mice of the CD1 strain (both males and females), a strain known to exhibit pathologies associated with inflammation at the greatest frequency among mouse strains used as experimental models in the laboratory [44], express serum levels of the proinflammatory cytokines TGF-β1, LCN2, and CXCL1, the murine equivalent of human IL-8, greater than those expressed by the serum from mice which instead more rarely develop inflammatory diseases in the protected environment of an animal facility such as C57BL6 or DBA2. These results indicate that the baseline serum concentration may represent a confounding factor when establishing the normal range of proinflammatory cytokines in a mouse strain and raise a cautionary note on the range of their concentrations, which should be considered normal in the human population, which also show a great baseline genetic heterogeneity.

The great level of proinflammatory cytokines present in the sera of CD1 mice with respect to those of other strains was associated with histopathological alterations in multiple organs predictive of pathologies associated with chronic inflammation (interstitial pneumonia, arteritis, glomerulo-nephritis, amyloidosis in the spleen, and epidermis hyperplasia). However, in over 60 old CD1 mice analyzed over the course of the years, we never observed myelofibrosis in the bone marrow (ARM, unpublished observation). Therefore, serum levels of proinflammatory cytokine, per se, do not predispose to the development of myelofibrosis in the absence of a driver mutation. This concept was further tested by determining that there is little difference in serum levels of a large panel (32) of proinflammatory cytokines, including TGF-β1, LCN2, and CXCL1, between *Gata1*^low^ mice and their wild-type littermates. There was also little difference between the serum levels of these cytokines between young, which do not express myelofibrosis, and old, when mice have developed myelofibrosis, *Gata1*^low^ mice. By contrast, the levels of TGF-β1, LCN2, and CXCL1 (and of BMP-4) were found greatly increased in the bone marrow of *Gata1*^low^ mice with respect to their wild-type littermates. Therefore, once leading mutations have appeared, the microenvironmental bioavailability of proinflammatory cytokine is of great importance for the development of myelofibrosis.

TGF-β1, LCN2, and CXCL1 are known to be expressed by multiple cell types. Our histochemistry studies indicated that they were expressed at comparable levels in most of the wild-type and *Gata1*^low^ microenvironmental cells analyzed, apart from megakaryocytes in which they were all expressed at levels greater than normal. These observations reinforce the knowledge that the malignant megakaryocytes play a major role in the development of myelofibrosis. Up to now, it was believed that this role was limited to secreting cytokines responsible to activate other cells (fibroblasts, monocytes-derived fibrocytes, or even other megakaryocytes) to synthesize proteins of the extracellular matrix, inducing fibrosis. Whether fibrosis is then a bystanding effect or is responsible for hematopoietic failure in the marrow is a matter of debate. The observation that myelofibrosis can be cured by bone marrow transplantation [73] provides a strong clinical indication that normal hematopoietic stem cells are capable of exerting their functions even in the altered microenvironment of a myelofibrosis patient, suggesting that fibrosis, per se, does not reduce the supporting HSC niches in the bone marrow of these patients.

The last few years have seen a great increase in our understanding of the niches supporting hematopoiesis in the bone marrow. The HSC niches in the murine bone marrow are defined based on their anatomical location (in the diaphysis or in the epiphysis of the femur), the type of blood vessels (sinusoids, arterioles of transition zone vessels) they contain, and the proximity with the endosteal zone [45,46,47]. Based on their anatomical site, at least two different types of HSC have been recognized: the central niche, which contains sinusoids and arterioles and hosts the majority (>90%) of the HSC, and the endosteal niche, in close proximity to the bone surface, which hosts the rest of the HSC. In both locations, HSC resides in close proximity to endothelial and mesenchymal stem cells [74]. It was Dr. Lord that first proved that the long-term repopulating HSC are localized in the most external niches of the bones, within niches going toward the central area of the medulla hosting progressively more differentiated HSC and finally progenitor cells [75]. Recent data indicate that the HSC niches are reshaped during aging and inflammation, and these changes are thought to determine the changes of HSC functions observed with age (HSC bias, exhaustion, and clonality) [72,76]. However, although it is well known that, in addition to microenvironmental clue, the effects exerted by aging on the property of HSC have a strong strain-specificity linked to the polymorphism of the polycomb gene among laboratory strains [77,78], our knowledge on the changes in the HSC niche with age in different strains is limited. By contrast with the human bone marrow, the mouse bone marrow contains very few adipocytes. It is debated whether these cells derive from mesenchymal stem cells or have HSC origin [79]. Whatever their origin is, by contrast with the adipocytes found in other tissues, the ones present in the bone marrow are capable of producing SCF and other HSC supportive growth factors [80] and have been proposed to be putative HSC niche, which becomes relevant with age. In agreement with this hypothesis, in old wild-type mice of the CD1 background, the small cells expressing the greatest levels of GFP were found localized in clusters surrounding the adipocytes. These results suggest that in this strain, adipocytes play important niche functions with age.

The bone marrow from *Gata1*^low^ mice contained number of adipocytes greater than that of their wild-type littermates. This observation is consistent with the notion that, in addition to fibrosis, the bone marrow from MF patients often contains great numbers of adipocytes, numbers that may be high enough to lead to the identification of a MF subtype, MF with fatty bone marrow, the etiology of which is still poorly understood [81]. It is possible that the driver mutations increase the proliferation potential to those adipocytes derived from the malignant HSC. In spite of their great number, adipocytes were seldomly associated with small cells expressing GFP in the bone marrow of *Gata1*^low^ mice. GFP+ cells were instead located in areas of the medulla surrounded by microvessels and bones and containing great numbers of megakaryocytes. These results suggest that the abnormal megakaryocytes increase the niches supporting short-term HSC and progenitor cells in the bone marrow. The hypothesis that the hematopoietic niches in an organ which is experiencing hematopoietic failure are increased is counterintuitive. This apparent paradox is partially explained by the observation that the bone marrows from *Gata1*^low^ mice contain great numbers of macrophages containing high levels of GFP in their cytoplasm. Since proinflammatory cytokines are well known to increase the apoptotic rates of HSC/progenitor cells [66], we suggest that megakaryocytes induce hematopoietic failure by secreting factors inducing HSC in apoptosis which are then phagocytized by macrophages.

## 5. Conclusions

An important conclusion from this study is that the resident self-tissue, also defined as microenvironmental bioavailability, of BM proinflammatory cytokine, rather than their systemic levels, plays a key role in the development of myelofibrosis in the *Gata1*^low^ mouse model. The major cytokines/chemokines identified are TGF-β1, LCN2, and CXCL1, and emperipolesis seems to be one of the major drivers of LCN2 secretion in the microenvironment.

## Figures and Tables

**Figure 1 biomolecules-12-00234-f001:**
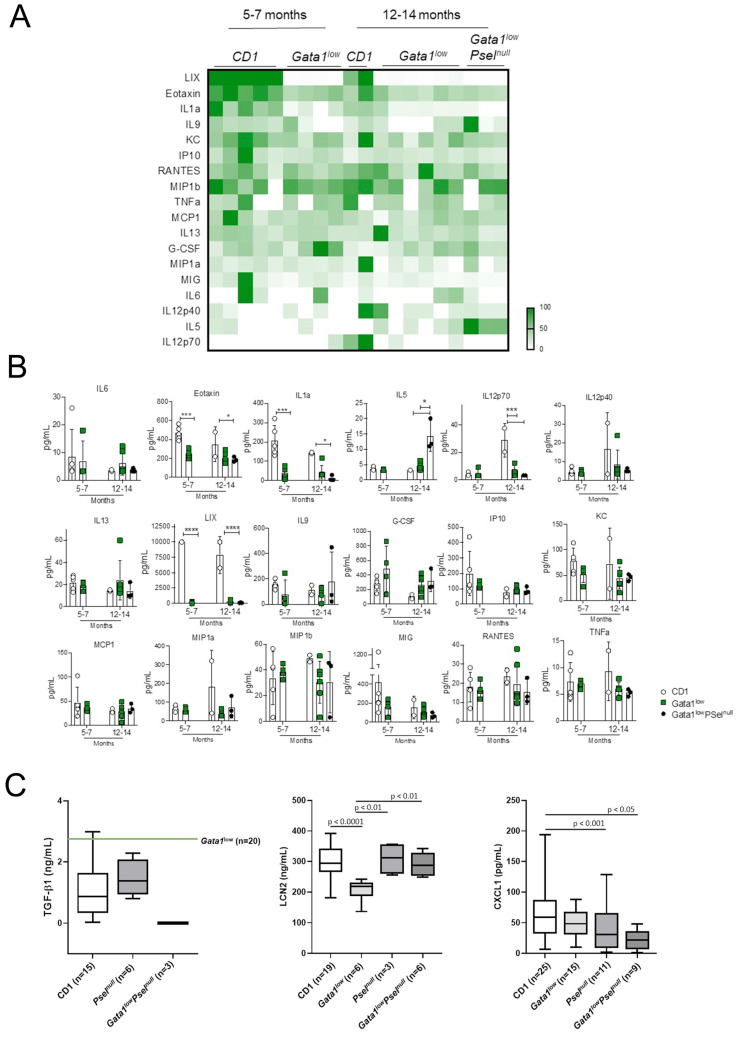
Modest changes in inflammatory cytokine profiling of sera from *Gata1*^low^ mice with age and with respect to CD1 littermates wild-type at the *Gata1* locus and to *Gata1*^low^*Psel*^null^ mice. (**A**) Heatmap of detectable cytokines. (**B**) Concentration values for each cytokine detected. *P* values were calculated with multiple comparison *t*-tests for each cytokine. * *p <* 0.05, *** *p <* 0.01, **** *p <* 0.0001. (**C**) Serum levels determined by ELISA of TGF-β1 (left), CXCL1 (middle), and LCN2 (right) in CD1 mice wild-type at the *Gata1* locus, in *Gata1*^low^ mice, and in mice lacking the *Psel* gene with and without the *Gata1*^low^ mutation. The number of mice analyzed in each experimental group is indicated by *n*. Statistical analyses were performed with multiple comparison test. Since age and sex analyzed as independent variables did not reveal statistically significant differences in most of the groups (see Appendix A), results obtained at different ages in females and males were pooled. The number of mice analyzed in each group was TGF-β1: adult CD1 males *n* = 5; adult CD1 females *n* = 3, old CD1 males *n* = 4; old CD1 females = 3, total CD1 *n* = 15; adult *Psel*^null^ males: *n* = 3; adult *Psel*^null^ females *n* = 3, total *Psel*^null^
*n* = 6; old *Gata1*^low^*Psel*^null^ females *n* = 3, total *Gata1*^low^*Psel*^null^ = 3; LCN2: adult CD1 males *n* = 5; adult CD1 females = 4, old CD1 males *n* = 5; old CD1 females *n* = 5, total CD1 *n* = 19; old *Gata1*^low^ males *n* = 3, old *Gata1*^low^ females *n* = 3, total *Gata1*^low^
*n* = 6; old *Psel*^null^ females *n* = 3, total *Psel*^null^
*n* = 3; old *Gata1*^low^*Psel*^null^ males *n* = 3, old *Gata1*^low^*Psel*^null^ females *n* = 3, total *Gata1*^low^*Psel*^null^
*n* = 6; CXCL1: adult CD1 males *n* = 8; adult CD1 females *n* = 6, old CD1 males *n* = 6; old CD1 females = 5, total CD1 *n* = 25; adult *Gata1*^low^ males *n* = 3, adult *Gata1*^low^ females *n* = 3, old *Gata1*^low^ males *n* = 4, old *Gata1*^low^ females *n* = 5, total *Gata1*^low^
*n* = 15; adult *Psel*^null^ males: *n* = 5; adult *Psel*^null^ females *n* = 3, old *Psel*^null^ females *n* = 3, total *Psel*^null^
*n* = 11; adult *Gata1*^low^*Psel*^null^ males *n* = 3, old *Gata1*^low^*Psel*^null^ males *n* = 3, old *Gata1*^low^*Psel*^null^ females *n* = 3, total *Gata1*^low^*Psel*^null^
*n* = 9. Abbreviations: TGF-β1, transforming growth factor β1; LCN2, lipocalin-2; Psel, P-selectin; IL6, IL1a, IL5, IL12p70, IL12p40, IL13, and IL9, interleukin 6, 1a, 5, 12p70 and p40, 13 and 9, respectively; LIX, also known as CXCL5 (C-X-C motif chemokine 5); G-CSF, granulocyte-colony stimulating factor; KC, also known as CXCL1 (C-X-C motif chemokine 1); MCP1, monocyte chemoattractant 1; MIP1a and MIP1b, macrophage inflammatory protein a and b; MIG, also known as CXCL9; TNFa, tumor necrosis factors a.

**Figure 2 biomolecules-12-00234-f002:**
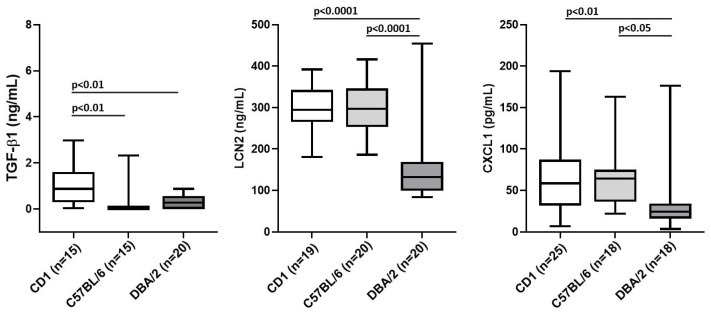
The serum from CD1 mice contains greater levels of proinflammatory cytokines than that from C57BL/6 or DBA/2 mice. The number of mice analyzed per experimental group is indicated by n. Since age and sex analyzed as independent variable did not reveal any statistically significant difference, results obtained at different ages in females and males were pooled (see Appendix A). The number of mice analyzed in each group was: TGF-β1: adult CD1 males *n* = 5; adult CD1females *n* = 3, old CD1 males *n* = 4; old CD1 females *n* = 3, total CD1 *n* = 15; adult C57BL/6 males *n* = 4; adult C57BL/6 females *n* = 3, old C57BL/6 males *n* = 5; old C57BL/6 females *n* = 3, total C57BL/6 *n* = 15; adult DBA/2 males *n* = 5; adult DBA/2 females *n* = 5, old DBA/2 males *n* = 5; old DBA/2 females *n* = 5, total DBA/2 *n* = 20; LCN2: adult CD1 males *n* = 5; adult CD1 females *n* = 4, old CD1 males *n* = 5; old CD1 females *n =* 5, total CD1 *n* = 19; adult C57BL/6 males *n* = 5; adult C57BL/6 females *n* = 5, old C57BL/6 males *n* = 5; old C57BL/6 females *n* = 5, total C57BL/6 *n* = 20; adult DBA/2 males *n* = 5; adult DBA/2 females *n* = 5, old DBA/2 males *n* = 5, old DBA/2 females *n* = 5; total DBA/2 *n* = 20; CXCL1: adult CD1 males *n* = 8; adult CD1 females *n* = 6, old CD1 males *n* = 6; old CD1 females *n* = 5, total CD1 *n* = 25; adult C57BL/6 males *n* = 3; adult C57BL/6 females *n* = 5, old C57BL/6 males *n* = 5; old C57BL/6 females *n* = 5, total C57BL/6 *n* = 18; adult DBA/2 males *n* = 5, adult DBA/2 females *n* = 4, old DBA/2 males *n* = 5; old DBA/2 females *n* = 4, total DBA/2 *n* = 18. *P* values were calculated with Tukey multiple comparison test and statistically significant differences (*p* < 0.05) indicated in the panels. Abbreviations: TGF-β1, transforming growth factor β1; LCN2, lipocalin-2.

**Figure 3 biomolecules-12-00234-f003:**
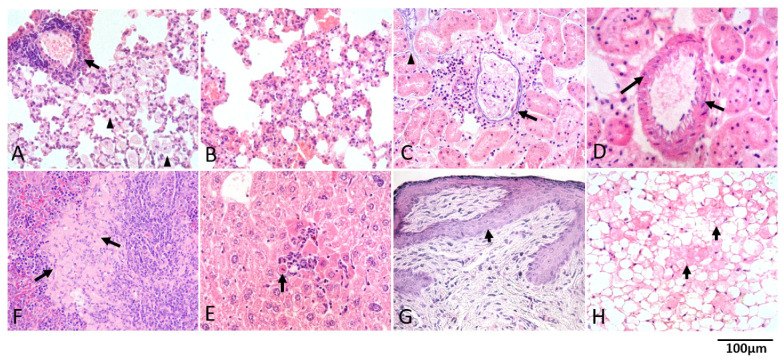
The organs from 12–14-month-old wild-type littermates of *Gata1*^low^ mice present a histopathological signature consistent with multisite inflammation. Hematoxylin/eosin staining of representative sections showing the presence of interstitial pneumonia with lymphocytes infiltrating the peri-broncho-vascular space (arrow), macrophages collection in alveoli (arrowheads) (**A**), and irregular thickening of alveolar septa by lymphocytes (**B**); Nephritis with a mixed expression of chronic glomerulonephritis (hypocellular glomerulus) and interstitial nephritis with lymphocytes infiltrating the inter-tubular and perivascular space with basophilic linear deposits along the basement membrane of Bowman’s capsule (arrow) and of tubules (arrowhead) (**C**); Arteriopathy with mural calcification: basophilic granular deposits beneath endothelial cells and between leiomyocytes, i.e., smooth muscle cells around the vessels (arrows) (**D**). Interstitial hepatitis with intralobular collection of lymphocytes surrounding shrunken and hyper-eosinophilic hepatocytes undergoing apoptosis (arrow) (**E**); Interstitial collection of faintly eosinophilic homogeneous material referable to amyloid (arrows) in the spleen (**F**); chronic dermatitis with dermal fibrosis, lack of adnexa, and hyperplasia of epidermis (arrow) with acanthosis in the skin (**G**), and steatonecrosis of the perirenal fat (arrows) (**H**). Magnification 40×. Results are representative of those observed in 15 mice (age ≥ 11 months, nine males and six females).

**Figure 4 biomolecules-12-00234-f004:**
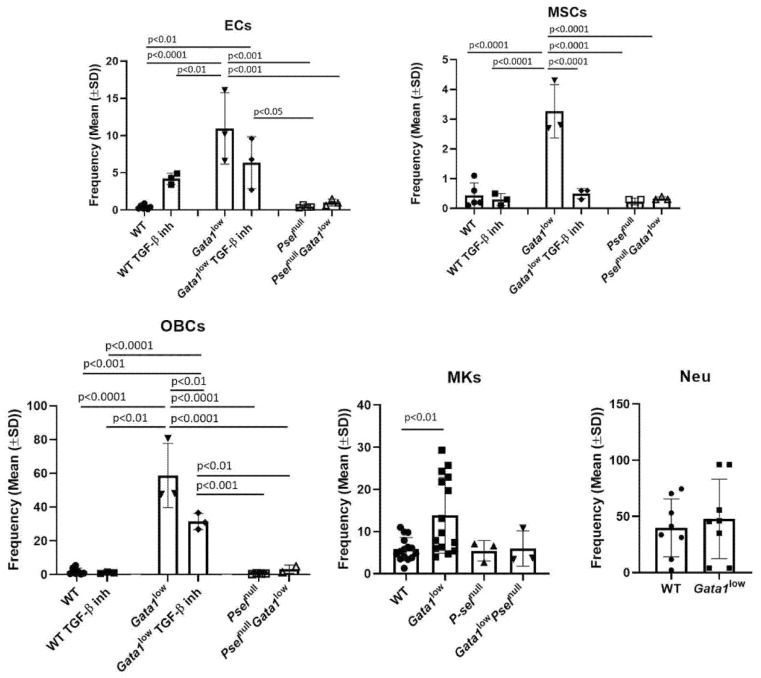
The bone marrow from *Gata1*^low^ mice (12–14 months of age) contains levels of endothelial cells, mesenchymal stem cells, osteoblasts, and megakaryocytes greater than those present in the bone marrow from wild-type littermates. The levels of these cells in the mutant mice were greatly reduced by treatment with a TGF-β1 inhibitor and/or by deletion of the *P-sel* gene. Cells were identified according to previously described flow cytometry criteria [34,35] (see Appendix A). Results are presented as mean (±SD) and as values of individual mice (each dot an individual mouse). *P* values were calculated with Tukey multiple comparison and those statistically significant are indicated in the panels. Abbreviations: ECs, endothelial cells; MSCs, mesenchymal stem cells; OBCs, osteoblasts; MKs, megakaryocytes; Neu, neutrophils; WT, wild-type; Psel, Pselectin; TGF-β inh, transforming growth factors β inhibitor.

**Figure 5 biomolecules-12-00234-f005:**
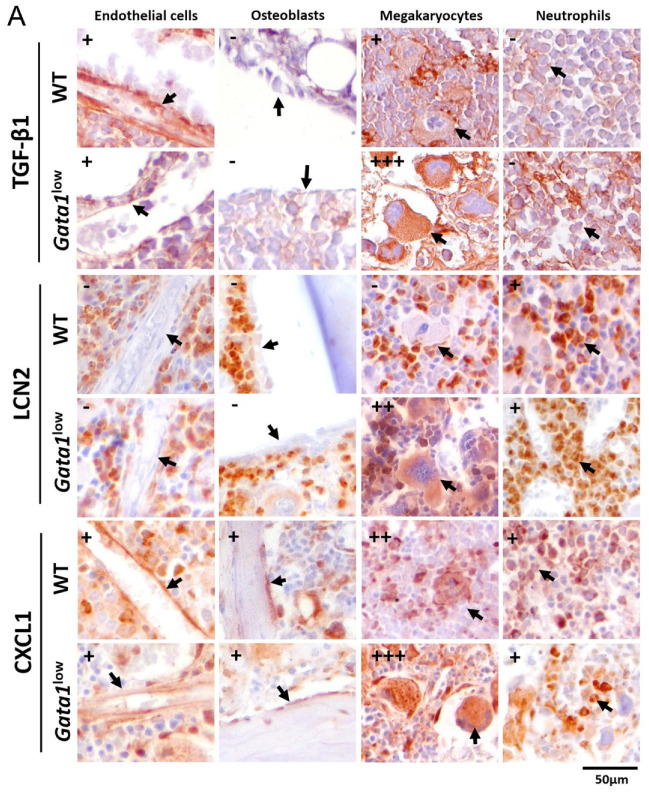
Megakaryocytes from the bone marrow of *Gata1*^low^ mice contain more TGF-β1, LCN2, and CXCL1 than the corresponding cells from the wild-type littermates. (**A**) Representative sections from the bone marrow of wild-type and *Gata1*^low^ littermates immunostained with antibodies per TGF-β1, LCN2, and CXCL1 showing the level of these cytokines in endothelial cells, osteoblasts, megakaryocytes, and neutrophils (all indicated by arrows) recognized by morphological criteria as indicated. Semiquantitative estimates (−, +, ++, or +++) of the intensity of the staining in each population is indicated on the top left. Original magnification 40×. Representative sections stained only with the primary antibody are shown in Appendix A) as negative controls. (**B**) Quantification of TGF-β1, LCN2, and CXCL1 expressing MKs of wild-type and *Gata1*^low^ mice. Frequency of total megakaryocytes, levels of fibrosis (as control), and percent of megakaryocytes expressing high levels of TGF-β1, LCN2, and CXCL1 in bone marrow section from *Gata1*^low^ and wild-type littermates, as indicated. The content of proinflammatory cytokines in the megakaryocytes was quantified as described in Appendix A) using five randomly selected areas per femur from three–four mice per experimental group. Statistical analysis was performed by *t*-test, and statistically significant p values among groups are indicated within the panels. Abbreviations: TGF-β1: transforming growth factor β1; LCN2: lipocalin-2; WT: wild-type; MKs: megakaryocytes.

**Figure 6 biomolecules-12-00234-f006:**
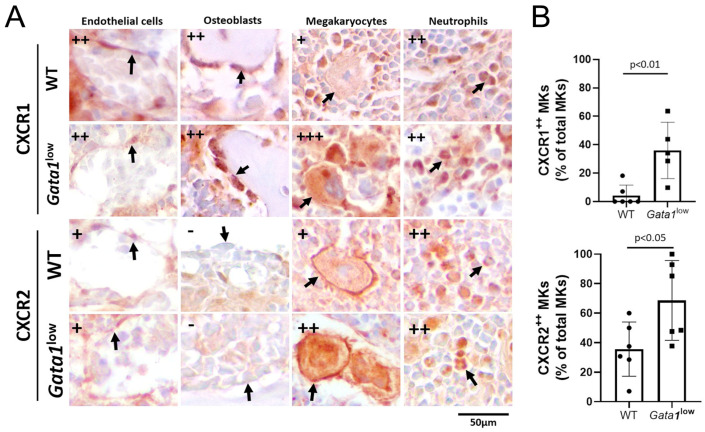
Malignant megakaryocytes express great levels of CXCR1 and CXCR2. (**A**) Representative sections from the bone marrow of wild-type and *Gata1*^low^ littermates immunostained with antibodies per CXCR1 and CXCR2, showing the level of these receptors in endothelial cells, osteoblasts, megakaryocytes, and neutrophils (indicated with arrows). Semiquantitative estimates (−, +, or ++) of the intensity of the staining in the different cells are provided on the top left. Original magnification 40×. Representative sections stained only with the secondary antibodies, are shown in Appendix A), as negative controls. (**B**) Percentage of megakaryocytes expressing high levels of CXCR1 and CXCR2. The proinflammatory cytokines were quantified as described in Appendix A) on five randomly selected areas per femur from three–four mice per experimental group. Statistical analysis was performed by *t*-test, and statistically significant p values among groups are indicated within the panels. Abbreviations: CXCR1 and 2: CXC motif chemokine receptor 1 and 2; WT: wild-type; MKs: megakaryocytes.

**Figure 7 biomolecules-12-00234-f007:**
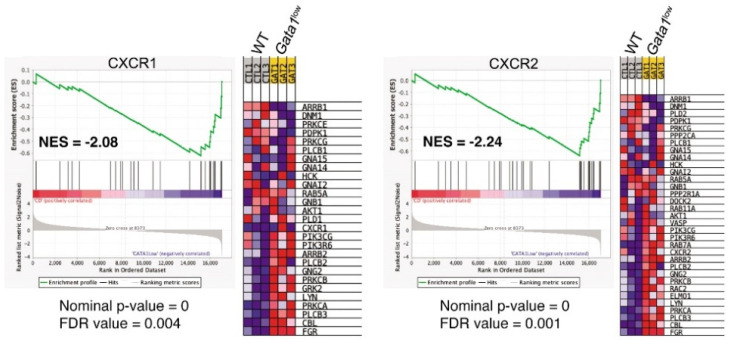
Both CXCR1 and CXCR2 gene signatures are enriched in *Gata1*^low^ bone marrow. Gene set enrichment analysis (GSEA) for the CXCR1 and CXCR2 signatures and heat maps of differentially expressed genes of the CXCR1 and CXCR2 pathway. Abbreviations: CXCR1 and 2: CXC motif chemokine receptor 1 and 2; WT: wild-type; MKs: megakaryocytes; FDR: false discovery rate.

**Figure 8 biomolecules-12-00234-f008:**
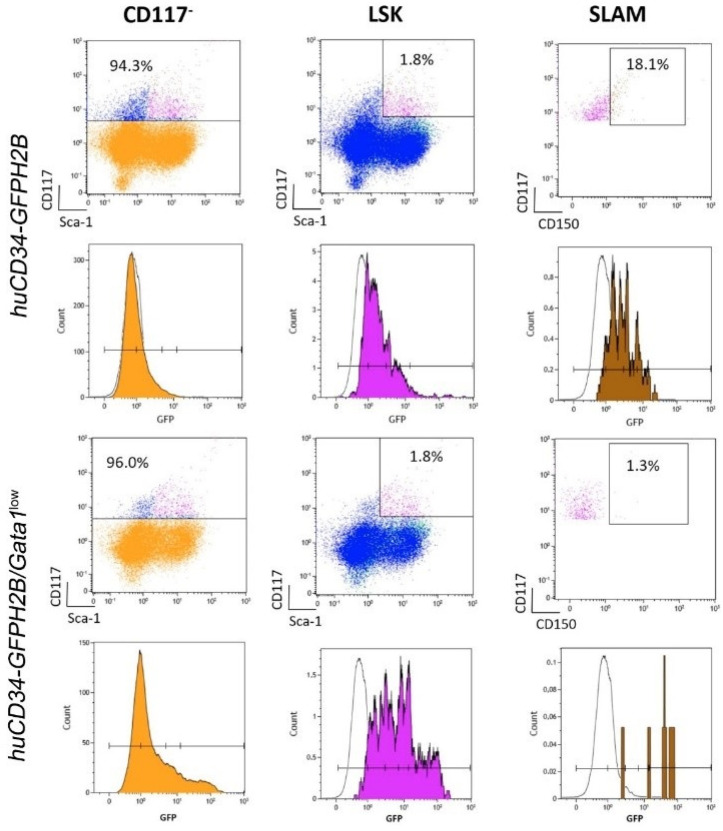
By flow cytometry, in the bone marrow of wild-type mice, GFP is expressed only by cells in the LSK gate, while in that from *Gata1*^low^ mice, GFP is expressed both by non-LSK and LSC cells. Scatter plots of bone marrow cells from representative *huCD34-GFPH2B* and *huCD34-GFPH2B/Gata1*^low^ littermates showing the gates used to identify cKIT^−^ cells (orange events), LSK (purple events), and SLAM (brown events) cells, as indicated. The intensity of the GFP expressed in the different populations is shown as color-coded histograms below each panel. The dotted histogram included in each panel indicates the signal in the GFP channels from the corresponding cells from the bone marrow of *hCD34* and *hCD34/Gata1*^low^ mice, used as negative controls. Abbreviations: CD117: cluster of differentiation 117 which recognize cKIT, the receptor for stem cell factor; LSK: cells that do not express lineage markers but express cKIT and Sca1; SLAM: LSK which also express markers of the signaling lymphocyte activation molecules (SLAM) family.

**Figure 9 biomolecules-12-00234-f009:**
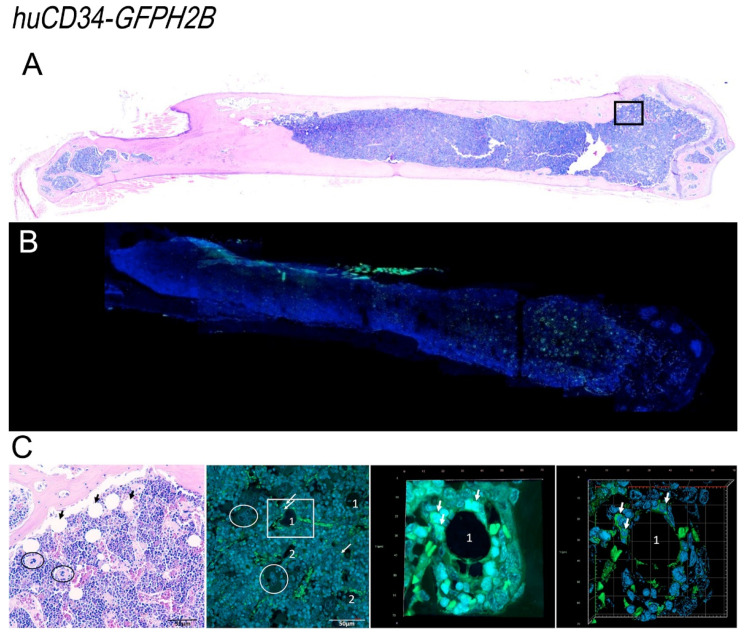
Confocal microscopy analyses of the distribution of GFP-positive cells within the bone marrow architecture of *huCD34-GFPH2B* mice. (**A**) The H&E staining of a femur section from one representative *huCD34-GFPH2B* mouse reveals hypercellular bone marrow with scant residual adipocytes well visible in H&E figure in C. (**B**) Confocal microscopy analyses with GFP and DAPI of a femur from one representative *huCD34-GFPH2B* mouse revealing that GFP+ cells are mostly distributed in the epiphysis of the femur. (**C**) Representative panels showing, at larger magnification, the areas indicated in rectangles in A (H&E staining, left panel) and in B (confocal microscopy, second panel of the left). The rectangle on the second panel on the left is shown as stack image and as tridimensional reconstruction on the right. The tridimensional reconstruction is shown in detail in movie 3. Black arrows in A indicate adipocytes, circles indicate megakaryocyte clusters, and white arrows indicate GFP-positive cells. A and B are a photo-merge of 4× magnification pictures; in C, the magnifications are 20× for the first and second panel, and 40× for the third and fourth panel.

**Figure 10 biomolecules-12-00234-f010:**
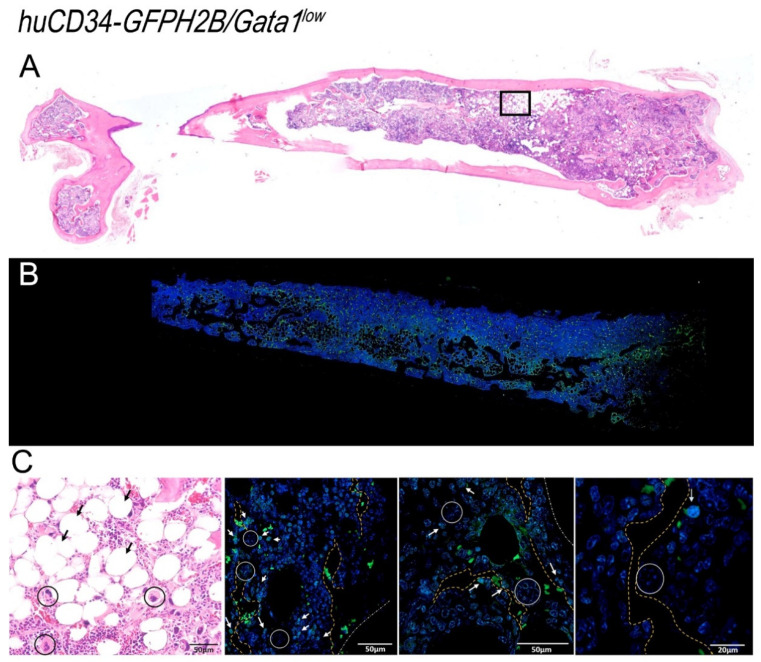
Confocal microscopy analyses of the distribution of GFP-positive HSC within the bone marrow architecture of *huCD34-GFPH2B/Gata1*^low^ mice. (**A**) H&E staining of a femur section from a representative *huCD34-GFPH2B/Gata1*^low^ reveals hypocellular bone marrow with adipocyte hyperplasia associated with fibrosis and abundance of megakaryocytes. (**B**) Confocal microscopy analyses with GFP and DAPI of a femur section from one representative *huCD34-GFPH2B* mouse revealing that GFP+ cells are distributed both in the epiphysis and in the diaphysis of the femur. (**C**) Left panel: larger magnification of the rectangle in the H&E-stained section in A showing the great numbers of adipocytes (black arrows) and megakaryocytes (circles) present in the femur of the mutant mice. Other panels: larger magnifications of the confocal microscopy analyses of the femur shown in B indicating the localization of the GFP+ cells within the bone marrow architecture. White arrows indicate GFP-positive cells, circles indicate megakaryocyte clusters, and white dashed lines indicate the contour of microvessels, while yellow dashed lines indicate the contour of the neoformed bones. A and B are a photo-merge of 4× magnification pictures; in C, the magnifications are 20× for the first and second panel, 40× for the third panel, and 60× for the fourth panel.

**Figure 11 biomolecules-12-00234-f011:**
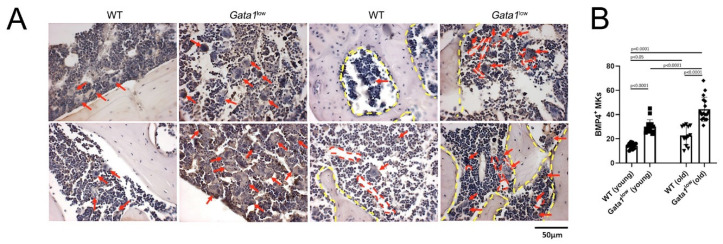
The megakaryocytes in the femur of *Gata1*^low^ mice contain high levels of BMP-4 and are localized in close proximity to microvessels and bone trabeculae. (**A**) Representative sections of the femur from two representative wild-type and *Gata1*^low^ mice immunostained for BMP-4 and counterstained with hematoxylin. Arrows indicate megakaryocytes, red dashed lines indicate the contour of the microvessels, and yellow dashed lines indicate the contour of the bones. Magnification 40×. (**B**) Quantification of the megakaryocytes expressing BMP-4 in the bone marrow of young and old wild-type and *Gata1*^low^ littermates, as indicated. Results are presented as mean (±SD) and as values of individual mice (each dot an individual mouse). *p* values were calculated with Tukey multiple comparison, and those statistically significant are indicated in the panels. Abbreviations: WT: wild-type; MKs: megakaryocytes, BMP-4; bone morphogenetic protein 4.

## Data Availability

Microarray data have been deposited in the Gene Expression Omnibus database (GSE89630) https://www.ncbi.nlm.nih.gov/geo/query/acc.cgi?token=ijgpwgsyjhwpdoj&acc=GSE89630.

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
