# Peer review of "Resident Self-Tissue of Proinflammatory Cytokines Rather Than Their Systemic Levels Correlates with Development of Myelofibrosis in Gata1low Mice"

_biomolecules, 2022, doi:10.3390/biom12020234_

Round 1

Reviewer 1 Report

In this highly interesting study, Zingariello M et al. clarify the relationships between the serum levels of pro-inflammatory cytokines, their bioavailability in the BM microenvironment, and their relevance for myelofibrosis development, using the Gata1low mouse model of myelofibrosis developed by the authors. In addition, they investigate the cellular source of key pro-inflammatory cytokines (mainly, TGF-β1, LCN2 and CXCL1 produced by malignant MKs). Furthermore, by using huCD34-GFPH2B/Gata1low mice, the authors show differences regarding HSC localization in Gata1low mice compared with WT mice. Interestingly, in the Gata1low BM, HSCs are located in the diaphysis in areas surrounded by micro-vessels and MKs, while WT HSC are found mainly located in the epiphysis, close to adipocytes.

This data-rich study provides strong arguments for the need to analyze the local bioavailability (rather than serum levels) of specific BM pro-inflammatory cytokines/chemokines, as key drivers BM fibrosis in MPN.  Below are minor comments aimed at further improving the manuscript’s presentation for publication.

  1. The authors showe in Figure 5B that LCN2 is mainly produced by BM neutrophils from WT mice, while in BM of Gata1lowmice, LCN2 is expressed at detectable levels both by neutrophils and MKs. The authors conclude that increased bioavailability of LCN2 in the microenvironment of Gata1low mice  is due to production by a novel malignant MKs. Since neutrophils are interact with MKs (emperipolesis) in MF, is it possible that LCN2 expression in Gata1low MK is associated/secreted from engulfed neutrophils in MK? This could be an interesting point for discussion.
  2. High CXCR1 and CXCR2 expression in the BM sections of Gata1lowmice is very interesting. Do the authors know which cells upregulate these receptors? This could provide more clarity and significance to the findings.
  3. hCD34 results: The interesting results described in lines 531-535 do not seem to be included in the figure. Either include or describe as ‘data not shown’ to avoid confusion. It might be worth clarifying that the hCD34 promoter also labels endothelial cells; there seem to be labelled blood vessels resembling arterioles, particularly in the Gata1low This might be an important finding because the authors describe the presence of GFP+ cells different from LSK cells in the Gata1low model. These cells seem to be Sca1+ by flow cytometry, and Sca1 is a marker of arterioles in the BM. Clarifying this point and accompanying the scatter plots of Figure 8 with quantification of individual experiments would substantiate the relevance and reproducibility of these findings.
  4. The relationships between LCN2, CXCL1 and the results presented in Figure 7 are not fully clear. Perhaps the paragraph in lines 453-462 could be reworded to clarify the line of thought.
  5. An important conclusion from this study is that the microenvironmental bioavailability of BM pro-inflammatory cytokine plays a key role in the development of myelofibrosis. The major cytokines/chemokines identified are TGF-β1, LCN2 and CXCL1, and emperipolesis seems to be one of the major drivers of LCN2 secretion. Perhaps these findings could be highlighted and discussed further in the Discussion.
  6. The reference (1016/j.immuni.2018.09.018) could be added to line 77, as evidence of murine IL-8 homologues MIP-2 (CXCL2) mainly produced by neutrophils.
  7. Figure 1C: does it make any difference to split the mice before or after disease onset?
  8. It would better to indicate cell numbers in the y axis of Figure 4. Line 439: typo “expressed al “should be “expressed at”. The section number in line 467 should be 3.6

Reviewer 2 Report

Dear Editor

First of all, I want to congratulate the authors for the great research work done.

The manuscript shows a large amount of data, which attempts to adequately answer the questions suggested by the authors.

The authors have experience in the proposed research study, with several publications related to the topic. The senior author (Dra. Migliaccio) in a recently published review, which largely discusses the results presented in this original article under review (PMID: 34943811).

From my perspective, the extensive work proposed by the research group is fully publishable.

As a reviewer of the manuscript, I would like to mention some details that I think may help the authors to get a better paper for the final publication.

Although it is unusual for reviewers to propose modifications to the title of the paper, given the complexity for authors to develop an attractive and encompassing idea for their original work, I would like to contribute to conceptualise a proposal for the paper based on what has been described in the literature.

The pro-inflammatory milieu is an increasingly evidenced pathophysiological concept and has been proposed as key mechanisms of damage in several pathological entities such as tissue manifestations of metabolic disorders such as nephropathy and diabetic retinopathy, atherosclerosis, cancer, among others.

It can be described as a local (resident self-tissue) or systemic (extravasation of pro-inflammatory elements into the bloodstream) inflammatory state.

The investigators' proposal consolidates just this idea and given the results observed in the pathophysiology of the disease (multi-organ inflammation with mild serum manifestation), the concept of local vs. systemic microinflammatory state could be used as keywords for the article under review. I hope the authors will take this suggestion as broad-mindedly as possible.

Concerning the content of the manuscript, some considerations:

Mayor Revisions

- There is a considerable difference between the numbers of animals proposed for each of the study groups (Figure 1C), even with a completely insufficient group of animals to draw statistically comparable conclusions (n=17 vs n=3). Additionally, the way of expressing the results of the Gata1low group is different in the same figure (TGF-B1 vs LCN2/CXCL1). It is suggested to unify criteria in graphs and to evaluate increasing n in group n=3.

- It is surprising that the authors use n = 25 in some groups. Increasing the number of animals does not mean greater research impact. The use of such a large number of animals needs to be adequately justified. In the very near future, this type of study could be invalidated just because it does not meet the criteria of animal care and experimentation. Although each investigation is unique, the use of 8-12 animals per experimental group is largely validated to obtain statistically significant conclusions.

- The argument mentioned by the authors (line 119-121) for mixing animals of the same sex is not an adequate approach for pooling or separating the results obtained. Given the high number of animals proposed (groups with n = 25) and that no gender differences are observed between the study groups, it is necessary for the authors to emphasise this hypothesis more strongly. Similarities and differences in gender studies are becoming increasingly important in preclinical research.

- Given the inflammatory component of immunohistochemistry, I suggest the authors add a negative control to the technique (without primary antibody), either in the figure or as supplementary material, similar to the reticulin analysis. This detail is very important, since beyond the authors' expertise in performing immunohistochemistry, readers should be aware of the selectivity of the proposed antibody.

- To answer the proposed hypothesis... Is it really necessary for the article to contain eleven figures? Although this is a more specific issue for the editorial team, the mention of figures with a single panel and the revalidation of the hypothesis with different techniques allow us to propose that certain figures be unified. This comment is a suggestion to the authors. 

Minor revisions

Figures should have abbreviations at the end of each caption, and I suggest that the number of animals in each group be displayed in this section rather than in the figure panel (Figure 1-2).

It is necessary to unify the criteria regarding the quality of images. There are images of very poor quality (blurred, loss of focus and margins), while others are extremely high quality (light, contrast).
